# Piloting a Nurse-Led Critical Care Outreach Service to Pre-Empt Medical Emergency Team Calls and Facilitate Staff Learning

**DOI:** 10.3390/ijerph20054214

**Published:** 2023-02-27

**Authors:** Anja Geisler, Susanne Hedegaard, Tracey K. Bucknall

**Affiliations:** 1Department of Anesthesiology, Zealand University Hospital, Lykkebaekvej 1, 4600 Koege, Denmark; 2Department of Clinical Medicine, University of Copenhagen, 2200 Copenhagen, Denmark; 3School of Nursing & Midwifery, Centre for Quality and Patient Safety Research, Institute for Health Transformation, Faculty of Health, Deakin University, 221 Burwood Highway, Burwood, VIC 3125, Australia; 4Centre for Quality and Patient Safety—Alfred Health Partnership, Institute of Health Transformation, Alfred Health, 55 Commercial Rd, Melbourne, VIC 3004, Australia

**Keywords:** clinical decision making, patient deterioration, rapid response system, nursing, critical care outreach, patient safety, medical emergency team, ICU liaison

## Abstract

A nurse-led critical care outreach service (NLCCOS) can support staff education and decision making in the wards, managing at-risk patients with ward nurses to avoid further deterioration. We aimed to investigate the characteristics of patients identified as at-risk, the types of treatments they required to prevent deterioration, the education initiated by the NLCCOS, and the perceived experiences of ward nurses. This prospective observational pilot study using mixed methods took place in one medical and one surgical ward at a university hospital in Denmark. Participants were patients nominated as at-risk by head nurses in each ward, the ward nurses, and nurses from the NLCCOS. In total, 100 patients were reviewed, 51 medical and 49 surgical patients, over a six-month period. Most patients (70%) visited by the NLCCOS had a compromised respiratory status, and ward nurses received teaching and advice regarding interventions. Sixty-one surveys were collected from ward nurses on their learning experience. Over 90% (*n* = 55) of nurses believed they had learned from, and were more confident with, managing patients following the experience. The main educational areas were respiratory therapy, invasive procedures, medications, and benefits of mobilization. Further research needs to measure the impact of the intervention on patient outcomes and MET call frequency over time in larger samples.

## 1. Introduction

Worldwide, rapid response systems (RRS) have been designed to prevent unexpected cardiac arrests and organ failure among hospitalized patients outside the intensive care unit (ICU) [1] and to respond to and manage deteriorating ward patients [2]. Although cardiac arrest rates have decreased since RRS have been put in place, in-hospital mortality results remain variable [2,3,4]. Since the introduction of rapid response teams, different constellations, led either by nurses or physicians, have been implemented to adapt to differing healthcare contexts [5,6].

RRS calls have primarily been based on respiratory challenges detected by changes in vital signs such as in the track-and-trigger algorithm, National Early Warning Score (NEWS) [7]. However, literature reporting the effectiveness of NEWS and RRS is of variable quality and in heterogeneous populations [8,9]. Assessing a patient’s degree of illness using scores based on routinely collected vital-sign measurements with pre-determined normality ranges remains insufficient to prevent all cardiac arrests and deaths among hospitalized patients [10]. Research has shown that signs of clinical deterioration, such as abnormal heart rate, respiratory rate, or oxygen saturation can occur several hours before serious adverse events [11,12]. Therefore, earlier recognition of deterioration and escalation is crucial for ward nurses to gain time to stabilize patients and perhaps mitigate serious adverse events [13,14]. Several machine-learning algorithms have been developed to support staff in identifying patterns which can predict patients who may deteriorate in the future [10,15,16]. However, machine-learning solutions have been infrequently implemented in hospitals and were supposed to be an aid to clinical decision making, not an alternative to skilled clinical judgment. Other solutions need to be considered to support nurses’ decision making in the wards, combining track-and-trigger systems and nurses’ clinical judgment [9].

Therefore, a nurse-led critical care outreach service (NLCCOS) was established at Zealand University Hospital, Koege, Denmark, in December 2018 as an adjunct to the nurse-led RRS. A critical care outreach service (CCOS) is defined as a multidisciplinary approach to ensure safe, equitable, and quality care for all acutely unwell, critically ill, and recovering patients irrespective of location or pathway (The National Outreach Forum, 2014). Additionally, CCOSs have shown that they can improve patient quality and safety by supporting ward nurses’ education and improving ward-based knowledge and communication [17,18]. Therefore, we hypothesized that when ward nurses and the NLCCOS care for a patient defined as at-risk together, the NLCCOS can provide ward nurses with additional knowledge, tools, and skills to strengthen decision making and provide a better understanding of the future care for a complex patient, to mitigate the risk of deterioration.

### Aim

This study investigated four areas: (1). the characteristics of patients subjectively defined as at-risk by ward nurses and managers; (2). the types and frequency of treatments implemented by the NLCCOS; (3). the education given by the NLCCOS; and (4). ward nurses’ perceived experience of the service.

## 2. Methods

### 2.1. Design and Ethics

A mixed-methods triangulation design was used to capture information from different but complementary sources to better inform our understanding of the intervention impact [19]. A pragmatic thematic analysis was used to identify, analyze, and report patterns from the answers in the questionnaires [20].This pilot study was approved by The Danish Data Protection Agency (REG-111-2018) and performed according to the recommendations from the Declaration of Helsinki [21,22]. No randomization or changes were made in the patient’s care or treatment. Therefore, according to Danish law, no approval was required from the regional or national Committee on Health Research Ethics in Denmark. The manuscript follows the Strengthening The Reporting of Observational Studies in Epidemiology (STROBE) guideline [23].

### 2.2. Setting

The study was performed in one orthopedic (30 beds) and one medical (30 beds) ward at a large university hospital in Denmark from December 2018 to May 2019. The usual practice at the hospital in terms of detecting clinical deterioration in patients is for the hospital staff to use a validated predictive tool, the National Early Warning Score (NEWS) [7,24,25]. Hospital policy instructs clinical staff to collect NEWS information to assist nurses in deciding how often they should measure patient vital signs and if a doctor or MET should be called.

### 2.3. Participants

A convenience sample of 100 patients at risk of deterioration was enrolled. Patients could be rated at-risk if the ward nurse had difficulties stabilizing the patient, worried, or had other challenges regarding the patient’s physical status, regardless of the NEWS. Usually, it would be the most care-intensive, challenging patient on the ward that day but not a candidate for a MET call at the time. The patient’s name and the location were handed over by the medical and surgical ward managers to the ICU ward manager every morning, Monday to Friday, during their shared morning conference. Afterwards, the NLCCOS would leave the ICU and go and meet the ward nurse, discussing the patient’s status and possible escalation. The participating nurses in the NLCCOS were trained ICU nurses, usually part of the MET. The participating ward nurses were those nurses taking care of at-risk patients on that particular day. The included patients were those nominated as at-risk by the ward managers. Exclusion criteria were patients requiring MET or needing immediate transfer to the ICU. No patient approval was obtained, since the study only used raw data and no patient-identifying information; therefore, according to Danish law, no approval was needed.

### 2.4. Data Collection Tools

NLCCOS nurses collected data in a Case Report Form (CRF). Data included patients’ vital signs, such as respiratory rate, saturation, blood pressure, heart rate, consciousness, and temperature. Additionally, pain levels measured by a numeric rating scale (NRS), level of consciousness by the Alert, Voice, Pain, Unresponsive (AVPU) score, oxygen treatment, suction, use of positive expiratory pressure (PEP) or continuous positive airway pressure (CPAP), level of resuscitation, plans for a revisit from the NLCCOS (yes/no), and finally, the treatment plan made by the NLCCOS and the ward nurse together, was described in text. NEWS was not a part of the data collection.

Ward nurses were asked to answer a short questionnaire after the NLCCOS had assisted them. It contained both quantitative and qualitative questions. The questionnaire was pilot-tested on ten people, five from the medical and five from the surgical ward (academics, ward nurses, and staff managers), before the study began, to check for clarity and feasibility of the data collection tools. No changes were needed to clarify the meaning of the questions.

The following questions were included in the questionnaire:To what extent do you feel confident in making decisions about the treatments following the NLCCOS visit and the teaching they have provided? (none, sparse, somewhat, very much)Within which areas did you learn something from the NLCCOS visit (treatments, practice skills, collaboration with other staff around the patient, pathology, how to stabilize the patient, others, please describe in text)If you did not feel that any learning took place, please describe what you think could be the reasons for that in text.

The questionnaires were kept in the ward until the study had ended to ensure anonymization.

### 2.5. Data Collection Processes

When the NLCCOS entered the ward, the ward nurse described the patient’s background for hospitalization and current condition. Then they went to the patient’s room together, explaining to the patient, if possible, why the NLCCOS was there and what they were going to do. After that, they reviewed the patient together and performed any interventions (such as suction and intravenous insertion). The NLCCOS described the background for the actions, showing how to perform the intervention correctly. Then, the CRF was filled out by the NLCCOS. Before the NLCCOS left the ward, s/he handed out the questionnaire to the ward nurse, who voluntarily filled it out and lodged it in a secure box within the ward.

### 2.6. Data Analysis

No formal sample-size calculation was performed, since the study was descriptive and explorative in nature. Quantitative data were analyzed using SPSS software (SPSS Statistics package for windows, version 21, SPSS Inc., Chicago, IL, USA). Non-parametric data were presented as medians and interquartile ranges (IQR). Parametric data were reported as means and standard deviations (SD). Categorical data were presented as frequencies (*n*) or percentages (%). Qualitative data were analyzed using an inductive approach inspired by thematic analysis [20]. Two individual authors read the statements regarding learning and lack of learning provided by the ward nurses as part of the questionnaires several times and identified themes that could be related to clinical practice. The themes were discussed afterward and changed if necessary until a consensus was reached.

## 3. Results

In total, 100 patients at risk of deterioration were reviewed over six months, 51 patients from a medical and 49 patients from a surgical ward. Most patients (71%) had a compromised respiratory status and needed supplemental oxygen, 23% used positive expiratory pressure (PEP), and 29% used continuous positive airway pressure (CPAP). Three quarters (73%) were to receive full resuscitation, if required; however, 18% had limitations on medical treatment and 2% had not had the issue considered (Table 1).

We only analyzed the individual vital signs which in normal practice would be used in combination as trigger score changes in the NEWS algorithm [26,27]. Vital signs were collected by ward staff prior to NLCCOS arrival. A typical patient was on oxygen therapy, with an increased RR and decreased oxygen saturation. One third had a compromised conscious state. (Table 2) illustrates the vital signs. After one month, 40% had passed away.

Sixty-one surveys were collected from ward nurses regarding their learning experiences, twenty-seven from the surgical ward and thirty-four from the medical ward. More than 90% (*n* = 55) of the respondents believed they had learned something from the NLCCOS visits and felt more confident in caring for the patient afterward. From the thematic analysis, it appeared that the NLCCOS identified the main educational areas for preventative care needed by ward nurses: different kinds of respiratory therapy strategies, invasive procedures, medications, and mobilization. Table 3 illustrates the types of treatments administered and plans for future interventions, which appeared after the qualitative thematic analysis. Nurses from the wards indicated that they found the learning situation, bedside discussions, and reflections on care beneficial to their practice.

Thirteen nurses did not believe any learning took place during the NLCCOS visit to the ward. The leading causes were the following: the patient had improved; the ward nurse knew what to do in the current situation and, therefore, did not feel they needed the knowledge provided by the NLCCOS; the treatment of the patient had ended and was considered palliative; and the ward nurse was prevented from participating in the learning situation.

## 4. Discussion

In this study, we found most patients identified as at-risk had a compromised respiratory status. As a result, the main educational areas were different kinds of respiratory therapies, diagnostic and treatment procedures, medications, and the need for patient mobilization. Ward nurses’ responses reflected high satisfaction with the support and guidance provided by the NLCCOS. Four main areas will be discussed in the following sections: the type of support nurses need to assist their clinical decision making, gaps in practice, types of patients needing support, and feasibility and acceptability of the NLCCOS.

### 4.1. Type of Support Nurses Need to Assist Their Clinical Decision Making

When patients change from stable to unstable, decision making becomes more complex, balancing and evaluating often subtle changes with nursing interventions and medical treatments [28]. Nurses’ ability to perform clinical decision making involves integrating knowledge and practical skills with thinking ability, attitudes, and values. When a patient is deteriorating, decision making is often under duress, further complicating the process [28,29]. Nurses spend most of their time with patients and are responsible for their frequent and ongoing physiological assessments [30]. Although anyone can call the RRS, the main requesters are nurses, because recognizing and interpreting physiological abnormality is primarily a nursing responsibility. Nevertheless, the need for escalation is not consistently recognized by nurses, even though the criteria may be fulfilled (NEWS > 6), or nurses are concerned. It might result from decreased nursing practice experience, fear of nurses appearing less intelligent, or not knowing the calling criteria [18,28]. Another factor is the increasing workload, resulting in lack of time to spend with the patients, missing the possibility to detect signs of deterioration. Our findings indicated a primary challenge for nurses was how to optimize a patient’s respiratory status and the treatments affecting it (inhalations, diuretics, mobilizing, and positioning the patient the right way). McKenna et al. described a lack of situation awareness as a problem [31]. Situation awareness is a vital skill that contributes to high-quality patient care and safety outcomes [32]. Based on Emsley’s three-level model includes, it includes nurses’ perception of patient data, comprehension of patient situations based on clinical cues, and projection of a patient’s future status [33]. Nurses may focus on other elements without recognizing that the patient is at-risk; therefore, the deteriorating patient could be missed [31]. Nurses are required to observe acutely ill patients for changes in vital signs and assess patients’ instability in a timely manner, including knowledge and comprehension of the patient’s current state and past medical history in terms of performing sufficient care. A literature review by Avalos et al. identified key factors that influenced nurses’ situation awareness as the following: data visualization, fatigue, nurse experience, care situations, shifts, and team communication [34].

### 4.2. Gaps in Practice

Prompt identification of patients going from one clinical state to a worse clinical state is crucial to avoid serious adverse events and higher mortality rates [35]. For that purpose, vital signs are collected and used systematically in the wards to guide future assessment and treatment plans [27]. Clinical factors predicting the development of severe conditions, which can require higher levels of treatment such as transfer to the ICU, have been detected using both a single parameter and combinations, as in the NEWS algorithm [36]. RR is an important physiological parameter that is a sensitive indicator of serious illness [37,38]. A large cohort of deteriorating medical patients had a significantly higher respiratory rate among all the vital signs collected [37]. Unfortunately, RR is dependent on the accuracy of measurement, since it must include counting the RR for 60 s instead of making an estimate [39,40]. Another problem we experienced was a lack of collected RR values (50%), which is also confirmed by findings from other studies [41,42]. The reasons could be, as explained by Ansell et al., a lack of clinical resources, a lack of specialized training, and little appreciation of the value of this vital sign with regard to clinical deterioration [43]. Our study found a median RR of 20 for all patients, which, according to the NEWS algorithm, should not be anything the nurses should be worried about. However, when combined with patients having oxygen support (70%), as described by Andersen et al., the progression of clinical deterioration is 40.5 times higher with the combination of a high respiratory rate and the need for an increased FiO_2_ [44].

Wards employ many newly registered nurses. In a study which reviewed nursing education and the clinical readiness of new graduate nurses, researchers found the education of nurses did not always prepare them for the contextual challenges of caring for deteriorating ward patients [31]. Some essential elements in clinical decision making include nursing knowledge, experience, the context of the decision-making situation, knowledge of the patient, interpretation of assessments, and reflection. These elements can be challenging for newly graduated nurses and in wards where nurses have to care for many patients simultaneously [31]. To ensure the correct response to patient deterioration, prompt escalation can be taught using simulation training [45]. One study showed that using simulation improved nurses’ ability to transfer knowledge to clinical practice and, thereby, the ability to identify deteriorating patients and enhance patient care [46,47,48].

### 4.3. Types of Patients Needing Support

Deterioration and serious adverse events are shown to be preceded by deterioration in vital signs in 60–84% of all cases [44]. The early recognition and correction of physiological abnormalities can improve patient outcomes by reducing the incidence of AEs. A nurse’s ability to identify, interpret, and act on physiological abnormality is a fundamental requirement in AE prevention [30,49].

Patients identified as at-risk and attended to by the NLCCOS mostly had compromised respiratory status. Learning about and discussing respiratory management was often required. However, at times, other fundamental nursing skills appeared deficient, for example, mobilization, hydration and nutrition requirements. These findings indicate a need to prioritize fundamental care [50] and promote time for shared decision making in clinical practice to support patients in a preventive way [51,52]. Another group of patients who were considered to be at-risk by the ward nurses and managers seemed to be patients in the transition to end-of-life treatment, even though three-fourths of the patients in the study were registered to receive full treatment. As described in other studies, there seems to be a gap between performing advanced life support as a part of a patient’s deterioration and performing end-of-life treatment [53]. Since 40% of the patients had passed away after one month, we hope the visit from the NLCCOS could support discussions about future care options and clinical realities containing patients’ values, reducing unwanted, burdensome, or intensive care treatment.

### 4.4. Feasibility and Acceptability of the NLCCOS

We found the majority of the ward nurses responding to the questionnaire felt they had obtained knowledge from the NLCCOS and consequently felt more confident caring for the patients afterward. Therefore, in this single site setting, it indicates that the NLCCOS can be considered a feasible solution to assist ward nurses in identifying patients at-risk and to facilitate improved management through education and support. These findings are supported by Mitchelle et al. and Chaboyer et al. [54,55]. A systematic review found an 80% reduction in the mortality rate, reduced cardiac arrest calls, and a decrease in length of stay. Despite the low quality of the included studies, the study indicated an overall positive trend and improved patient care by introducing a NLCCOS [56]. The definition of “at-risk” used in this study design was not based solely on vital signs but also on the ward nurses’ gut feeling or feeling worried for the patient. A systematic review by Douw et al. [57] finds the signs and symptoms that trigger nurse worry were the following: change in respiration, change in circulation, rigors, change in mentation, agitation, pain, unexpected trajectory, patient indication they are feeling unwell, subjective nurse observation, and nurse being convinced something is wrong without a rationale. A visit from the NLCCOS provides an opportunity for the ward nurse to translate intuitive feelings into words and, thereby, through the discussion, supports the ward nurse’s decision making, empowering the nurse to find suitable actions for the situation.

### 4.5. Strengths and Limitations

This study used a convenience non-randomized patient cohort in only two ward settings in a university hospital in Denmark. As a pilot study, we showed the approach was feasible and acceptable, but we did not collect patient outcomes. Measuring patient outcomes of those patients receiving the NLCCOS review and support compared to wards that did not receive the additional support would offer further insight into the potential effectiveness of the intervention. Another limitation was the absence of a total NEWS for each patient assessed, and it is recommended to compare patient cohorts with the wider research evidence. The response rate of 61% for the ward nurses’ surveys was considered adequate. However, more in-depth interviews with ward nurses would have complemented the data collection. Further studies could incorporate this approach to provide greater insight into the educational needs of staff and the impact of targeted training and facilitation on the wards.

## 5. Conclusions

In this pilot study, we established the feasibility and acceptability of the NLCCOS in responding to patients identified as at risk of deterioration by ward nurses. We also identified common patient characteristics and treatments considered for a patient at-risk in the ward and the learning requirements of ward nurses to mitigate the risk of further deterioration. Furthermore, the study highlighted the perceived benefits of interactive bedside learning and reflective practice, especially for newly educated nurses. Opportunities for interactions between ward nurses and nurses from the ICU offer the potential for building relationships and confidence in caring for unstable ward patients. Accurate situation awareness is a critical component for nurses to detect deteriorating patients, but it requires training and focus to be developed in busy inpatient wards. Further research should test this model of care for effectiveness in a randomized trial.

## Figures and Tables

**Table 1 ijerph-20-04214-t001:** Characteristics and treatment of patients identified as at-risk (N = 100).

Characteristic	*n*
Gender, male/female/*	47/49/*
Respiratory rate, median/*	20/*
SAT, median	94
Oxygen, median, liters	2
Heart rate, median/*	91/*
Temperature, median	37
Pain NRS (0–10), median/*	1/*
Pain > NRS 3	16
Treatment, oxygen	71
Treatment, PEP	23
Treatment, CPAP	29
Treatment, suction	11
Level of resuscitation/* Full treatment CPR without ICU ICU without CPR Not documented	72 15 3 2
A new NLCCO visit scheduled	5

Abbreviations: PEP = positive expiratory pressure, CPAP = continuous positive airway pressure, NRS = numeric rating scale, NLCCOS = nurse-led critical care outreach service, sys = systolic, EWS = Early Warning Score, CPR = cardiopulmonary resuscitation, SAT = saturation, *n* = numbers, L = liter. * Missing data, totals may vary.

**Table 2 ijerph-20-04214-t002:** Patients’ vital signs (*n* = 100).

Vital Sign	n
Respiratory rate/n/*	
12–20	25
9–11	0
21–24	9
>24, <9	14
Saturation/%/*	
>95	11
94–95	11
92–93	8
<92	13
Oxygen/L/min/*	
No oxygen	28
Oxygen	71
Heart rate/bpm/*	
51–90	63
91–110	14
111–130	9
>130,<41	3
Blood pressure, systolic/*	
111–219	15
101–110	0
91–100	8
>219, <91	0
Temperature/°C/*	
36.1–38	70
38.1–39, 35.1–36	12
>39	0
<35	0
AVPU/*	
A	63
V,P,U	33

AVPU score = Alert, Voice, Pain, Unconcsious; * Missing data: totals will not add up to 100; bpm = beats per minute.

**Table 3 ijerph-20-04214-t003:** Interventions provided by the NLCCOS and examples of plans made for future care.

Themes	Interventions	Examples of NLCCOS Plans
Respiratory therapy	Suction	We will help you with the suction Expectorate sample
	CPAP *	Stop CPAP the patient is for palliation The patient needs CPAP please use the system You need to give the patient CPAP more frequently
	PEP *	Frequently PEP PEP 4–5 times/day I have told the nurse what a PEP flute is
	Oxygen	Give the patient a mask The patient need high-flow oxygen
Invasive procedures	ABS *	New ABS in 2 h. Take an ABS
	Gastric tube	The patient doesn’t eat much so needs a gastric tube A gastric tube is needed for nutrition and medication
	Other	The patient needs a urine catheter Take a sample from the secretions Blood test for fluid status
Mobilization		Remember frequently to turn the patient from side to side You must mobilize the patient to reduce the secretions Mobilize and position the patient to avoid obstruction The patient needs an air pulsation mattress
Medications	Diuretic	Consider diuretics to improve the respiratory status and to lower the blood pressure Pulmonary stasis, patient needs diuretics
	Pain	The patient are in pain he needs more analgesics A plan for pain is needed so the patient can sleep and you can avoid delirium
	Inhalations	Combivent is needed Sodium inhalations If respiratory aggravation then inhalations needed with sodium
Mixed group		Be aware that the patient has problems when swallowing Weigh the patient dailyThe patient needs palliation

* ABS = arterial blood sample. CPAP = continuous positive airway pressure. PEP = positive expiratory pressure.

## Data Availability

Data supporting reported results can be obtained by contacting Anja Geisler at: agei@regionsjaelland.dk.

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
