# Peer review of "Piloting a Nurse-Led Critical Care Outreach Service to Pre-Empt Medical Emergency Team Calls and Facilitate Staff Learning"

_ijerph, 2023, doi:10.3390/ijerph20054214_

Round 1
Reviewer 1 Report
Thank you for the opportunity to get acquainted with your interesting work. Although the topic is very interesting and current, I have doubts about the quality of the study:
1) What was the study endpoint? Only the results of the survey, or the actual intervention on patients?
2) Was there a change in the number of RRS calls before and after the training?
3) Were NLCCOS trainers specially selected?
4) No comparisons of the results of other studies in the discussion. There are numerous studies to refer to and cite, such as:
a) Klejne T, Jayamaha AR. Knowledge of the in-hospital resuscitation algorithm among medical staff of selected hospital departments. Crit. Care Innov. 2019; 2(2): 9-16.
b) Garry L, Rohan N, O'Connor T, Patton D, Moore Z. Do nurse-led critical care outreach services impact inpatient mortality rates? Nursing in Critical Care 2019; 24(1): 40-46.
5) The role of "support" for nurses should be further clarified. How do the authors propose to implement this assistance in wards?
Author Response
Reviewer: What was the study endpoint? Only the results of the survey, or the actual intervention on patients?
Authors: We have now tried to clarify the outcomes to “This study investigated the characteristics of patients subjectively defined as “at-risk” by ward nurses and managers, the types and frequency of initiated treatments, and the education performed by the NLCCOS, and the ward nurses´ perceived experience of the service”.
Reviewer: Was there a change in the number of RRS calls before and after the training?
Authors: The focus of the study was on the feasibility and acceptability of the NLCCOS role. We did not attempt to measure patient outcomes in this pilot study.
Reviewer: Were NLCCOS trainers specially selected?
Authors: The NLCCOS is comprised of trained ICU nurses, who are usually part of the MET. They were not specifically selected for the study.
4) No comparisons of the results of other studies in the discussion. There are numerous studies to refer to and cite, such as:
- a) Klejne T, Jayamaha AR. Knowledge of the in-hospital resuscitation algorithm among medical staff of selected hospital departments. Crit. Care Innov. 2019; 2(2): 9-16.
- b) Garry L, Rohan N, O'Connor T, Patton D, Moore Z. Do nurse-led critical care outreach services impact inpatient mortality rates? Nursing in Critical Care 2019; 24(1): 40-46.
Authors: Very good point. We have now added a new section in the discussion called “Feasibility and acceptability of the NLCCOS”.
5) The role of "support" for nurses should be further clarified. How do the authors propose to implement this assistance in wards?
Authors: The aim of the pilot was to test the feasibility and acceptability of the process at a single site before expanding the intervention to test the effectiveness on patient outcomes in a study that has sufficient power across multiple sites. The next step for the researchers will be to conduct a single site RCT, implementing the intervention across multiple wards to measure patient outcomes. Once this has been conducted, if improved outcomes are found then a larger multi-site RCT would be conducted. Rohan et al identified that more than half the studies in their systematic review were single site, with poor quality and low validity. To expand the service to multiple wards, a dedicated CCO nurse will be required to allow for greater time in the wards without be called as a member of the MET.
Reviewer 2 Report
Dear authors, this is a research paper with clinical importance for raising and upgrading the role of nursing care in the routinely hospital ward care. Please accept my following comments.
You mentioned that the Danish hospitals use the EWS and you wrote the physiological parameters with their values, limits and scores. What exaclty EWS are you describing (e.g. MEWS, NEWS or something else- a specific danish?) and if yes, what is the evidence (please give some references, about the validation of this specific “EWS”). Ref No. 22 and 25 are not for this specific EWS.
Usually after the assessment by EWS, there is a patient's clinical risk classification, followed by a clinical response (escalation of the care). In your study population, was any algorithm followed or the NLCCOS / ward nurses made their own decision making related to each case? This is a critical point in any EWS (to follow the recommendations in the field of clinical response after any EWS score).
Despite that you didn't measure any health outcome, also you haven't reported the EWS scores finally. This result has to be reported in order to know the severity and the clinical situation of the patients. It is not correct to report only values (table 2) without the total score of the EWS of the study population and some statistics about this (eg. pathological /surgical, etc.)
Generally speaking the paper raises some critical points liek the repsoratory assesment as a vital sign and the supporting role of the NLCCOS in daily ward care.
Best regards,
Author Response
Reviewer: You mentioned that the Danish hospitals use the EWS and you wrote the physiological parameters with their values, limits and scores. What exactly EWS are you describing (e.g. MEWS, NEWS or something else-a specific Danish?)
Authors: We have clarified that we use the NEWS and revised the manuscript accordingly.
Reviewer: if yes, what is the evidence (please give some references, about the validation of this specific “EWS”). Ref No. 22 and 25 are not for this specific EWS.
Authors: We have added new references in the text.
Reviewer: Usually after the assessment by EWS, there is a patient's clinical risk classification, followed by a clinical response (escalation of the care). In your study population, was any algorithm followed or the NLCCOS / ward nurses made their own decision making related to each case? This is a critical point in any EWS (to follow the recommendations in the field of clinical response after any EWS score).
Reviewer: Despite that you didn't measure any health outcome, also you haven't reported the EWS scores finally. This result has to be reported in order to know the severity and the clinical situation of the patients. It is not correct to report only values (table 2) without the total score of the EWS of the study population and some statistics about this (eg. pathological /surgical, etc.)
Authors: Only single parameters were collected in the CRF, these were transformed by the researchers to identify the severity of the patient symptoms if compared to individual NEWS. A total NEWS was not collected during in this study. This is a limitation of the study and mentioned in the limitations. For the next planned study, a total NEWS will be collected to compare results in more detail with the literature. I have also removed the summarized NEWS score from Table 2 to avoid misunderstandings.
Reviewer 3 Report
I considered that this manuscript has structural weaknesses in the methodology, specifically in the design of the methods. Given the way the objectives are elaborated, which should be substantially revised, it seems to me that this is a mixed methods design. The authors therefore should clarify and improve substantially the objectives and the design of the study.
Author Response
I considered that this manuscript has structural weaknesses in the methodology, specifically in the design of the methods. Given the way the objectives are elaborated, which should be substantially revised, it seems to me that this is a mixed methods design. The authors therefore should clarify and improve substantially the objectives and the design of the study.
Authors: We have reported multiple types of data separately and not integrated the findings only in the discussion paragraph. We have now added the following to the Methods paragraph and hope you find it satisfactory “A mixed methods triangulation design was used to capture information from different but complimentary sources to better inform our understanding of the intervention impact”.
Round 2
Reviewer 3 Report
Dear Authors,
I consider that the qualitative component, specifically when they refer to content analysis, should be better clarified.
In my point of view it should be specified what kind of analysis was performed, if deductive, inductive, or both; how the data were worked on, if any Computer-assisted qualitative data analysis software (CAQDAS) was used.
Author Response
Reviewer 3
I consider that the qualitative component, specifically when they refer to content analysis, should be better clarified.
Authors: We have added the following to the method paragraph “Qualitative data was analyzed using an inductive approach inspired by thematic analysis. Two individual authors read the statements regarding learning and lack of learning provided by the ward nurses as part of the questionnaires several times and performed themes that could be related to clinical practice”.
In my point of view it should be specified what kind of analysis was performed, if deductive, inductive, or both; how the data were worked on, if any Computer-assisted qualitative data analysis software (CAQDAS) was used.
Authors: We have now added inductive approach. We did not use any kind of analysis software for the qualitative analysis